The introduction of an invasive weed was not followed by the introduction of ethnobotanical knowledge: a review on the ethnobotany of Centaurea solstitialis L. (Asteraceae)

Branco Soraia 1
Irimia Ramona E. 1 2
Montesinos Daniel daniel.montesinos@jcu.edu.au 1 3
1 Centre for Functional Ecology, University of Coimbra , Coimbra , Portugal
2 Plant Evolutionary Ecology, Institute of Evolution and Ecology, University of Tübingen , Tübingen , Germany
3 Australian Tropical Herbarium, James Cook University , Cairns , Queensland , Australia
Heger Tina
Electronic publication date: 2023 Jun 7
Publication date: 2023
Volume: 11
Electronic Location ID: e15489
Received 2022 Aug 2; Accepted 2023 May 10
Copyright: ©2023 Branco et al.
Copyright year: 2023
Copyright holder: Branco et al.
License: This is an open access article distributed under the terms of the Creative Commons Attribution License, which permits unrestricted use, distribution, reproduction and adaptation in any medium and for any purpose provided that it is properly attributed. For attribution, the original author(s), title, publication source (PeerJ) and either DOI or URL of the article must be cited.
License URL: https://creativecommons.org/licenses/by/4.0/

Keywords: Biogeography, Ethnobotany, Ethnopharmacology, Invasive alien species, Yellow star-thistle

Funding: The authors received no funding for this work.

==============================
Invasive plants are known for their impacts to ecosystems and societies, but their potential cultural use tend to be unexplored. One important mechanism of plant invasion is the use of “allelochemicals” or “novel weapons”: chemical defenses which are new to their invaded habitats and that confer them competitive advantages. However, these chemicals are precisely what confers them ethnobotanical and medicinal properties. We reviewed the literature assessing the biogeography of the cultural uses of the model invasive plant yellow-starthistle (Centaurea solstitialis L.; Asteraceae), and assessed the extent to which the introduction of a weed native to Eurasia into several non-native world regions was paralleled by the spread of cultural uses from its native range. We found that the species was rich in pharmaceutically active compounds and that the species had been traditionally used for medicinal purposes, as raw material, and as food. However, ethnobotanical uses were reported almost exclusively in its native range, with no uses described for the non-native range, apart from honey production in California, Argentina, and Australia. Our study exemplifies how, when plant introductions are not paralleled synchronously by significant human migrations, cultural adoption can be extremely slow, even within the native range of the species. Invasive species can provide real-time insights into the cultural processes by which humans learn to use plants. This case study highlights how biological invasions and cultural expansions can be subjected to different constraints.

Introduction

With the intensification of globalization and trade, humans have intentionally or accidentally lead to the spread of alien invasive plants from one environment to another and, often, plants that were considered economically and ecologically valuable in their native regions became unwanted invaders in the introduced areas. Invasive weeds can wreak havoc in the non-native regions that they invade, but in their native range they are often valued medicinal plants or, alternatively, an inconvenient but not highly problematic native weed (Hierro, Maron & Callaway, 2005; Montesinos, 2022). The reasons why they are not problematic in their native range are multiple, including the presence of other plant competitors, herbivores, pathogens, and parasites that share a long evolutionary history with the weed (Callaway & Maron, 2006; Enders et al., 2020). An important factor involved in the disproportionate success of invasive plants in their non-native regions can be the use of “novel weapons” (Callaway et al., 2008). This term refers to the presence of plant chemical defenses that are new to the invaded plant communities, giving invasives a disproportionate success in their non-native ranges (Hierro & Callaway, 2003). However, in their native range, natural communities have been exposed to these chemicals for extended periods of time, allowing native communities to develop a tolerance to these chemicals (Schaffner et al., 2011; Becerra et al., 2018). These plant chemical compounds are precisely the ones responsible for the numerous ethnobotanical and medicinal uses that can frequently be found in the native ranges of these weeds. As such, non-native introduced plant species should be strong candidates for ethnobotanical adoption also in the ranges where they are introduced (Pfeiffer & Voeks, 2008; De Medeiros et al., 2011; Dos Santos et al., 2014; Gaoue et al., 2017) where they can be adopted as valuable medicinal plants (De Albuquerque, 2006; Pfeiffer & Voeks, 2008; Dos Santos et al., 2014; Maema, Potgieter & Samie, 2019), often in a balancing act between counteracting their environmental impacts, while benefiting from their medicinal or economic uses (Rakotoarisoa et al., 2016; Shackleton & Shackleton, 2018; Maldonado & Voeks, 2021). For instance, a study of medicinal uses of alien plants introduced into South America found that alien weeds can become an important component of the local pharmacopeias (Bennett & Prance, 2000); and another study found that in the Mexican region of Chiapas the proportion of alien weeds used for medicinal purposes was higher than should be expected given their relative abundance (Stepp & Moerman, 2001).

There is abundant research about the ecological impacts of invasive species, but not so much attention is given to the traditional use accumulated through the centuries in their original native ranges. Although highly controversial, some authors claim that one of the best strategies to control invasive species is through consumption (see review in Nuñez et al., 2012). For instance, in the native range of the weed Centaurea solstitialis L. the plant is traditionally fed to sheep (Kargioǧlu et al., 2008), and it has been proven that livestock grazing is an effective way to reduce the number of C. solstitialis flower heads with about 75% to 90% in the invaded region of California, where grazing has been used as a measure of biological control (Thomsen et al., 1993). Studying traditional uses could be of great importance to understand the idiosyncrasy of invasive species from their very origins, and thus to develop new research and management plans.

The goal of this study is two-fold. Firstly, we aimed to review and synthetize the ethnobotanical uses of an important global invasive weed. Secondly, we aimed to assess the geographical variation of those ethnobotanical uses across the native and non-native world regions where it is present. To achieve this, we reviewed the ethnobotanical literature available for the model invasive plant species, C. solstitialis L. (Asteraceae), native from Eurasia and invasive across the Americas and Australia, and compared it with any reference or reports of cultural and ethnobotanical use across the world regions in which it is considered invasive. Centaurea solstitialis is an annual forb adapted to disturbed environments (Grime, 1974; Xiao et al., 2016). Seeds of the species were introduced as a contaminant of agricultural seeds in many regions around the world over the last two centuries (Wang et al., 1991; Eriksen et al., 2014; Irimia et al., 2021). Although it is considered a noxious weed in most of its introduced range, in its native range this species has been subject to cultural experimentation locally, becoming an important element in the local culture and gastronomy (Guarrera & Lucia, 2007; Lentini & Venza, 2007; Farouji & Khodayari, 2016; Licata et al., 2016; Geraci et al., 2018), traditional medicine (Gunes, 2017), or as raw material (Kargioǧlu et al., 2008). Many studies have been carried out to identify the chemical compounds that make this plant pharmacologically relevant, and so far it is known that the species possesses many sesquiterpene lactones with a broad spectrum of biological activities (Özçelik et al., 2009), which are variable across the world (Irimia et al., 2019). Additionally, C. solstitialis is regarded as an important plant for honey production in California, (Zouhar, 2002).

This review aims to exemplify how the study of the ethnobotanical use of a model invasive species can provide important information about the biogeography and history of ethnobotany. We aim to summarize the available information about traditional uses, pharmacological activities, phytochemistry, and toxicological research available, to identify knowledge gaps, and to provide a scientific basis for potential applications in resource management. Finally, we aim to shed light on whether, and to what extent, ethnobotanical knowledge can be transmitted when an exotic species is introduced across different world regions.

Methodology

Model study species

Yellow star-thistle (Centaurea solstitialis L.; Asteraceae) is an erect winter annual weed (occasionally biennial), which usually grows up to 1 meter tall, sometimes up to 2 m tall, with spiny yellow-flowered heads (DiTomaso, 2001). Anatolia and the Caucasus are considered to be the ancestral range of the species (Eriksen et al., 2014), from where it went through a step wise range expansion into central and southern Europe which is nowadays regarded as adventitious or “expanded range” (Hierro et al., 2009). Several subspecies of C. solstitialis have been described throughout the native range, four in Europe (Garcia-Jacas et al., 2006) and three in the Asian part of Turkey. Starting in the mid-1800s, C. solstitialis was introduced as an agricultural seed contaminant in many regions around the world, including the western United States (USA), southern South America, southern Africa, and southern Australia (non-native range) (Hierro et al., 2016). The degree of invasive success is variable across the introduced range, with the species being highly damaging in Argentina and California (USA) (Hierro et al., 2011). Centaurea solstitialis is consistently diploid across its native and non-native ranges, and thus invasive success is attributed to other life history and ecological traits (Irimia et al., 2019). This plant is a major consumer of ground water and it costs the California state millions of dollars in water loss for wildlife, agriculture, and municipal uses. It was estimated that in the year 2004 the water lost from plants of C. solstitialis in the Sacramento River watershed costed between $16 million and $75 million dollars per year (calculated using the June 1999 CALFED cost estimates) (Gerlach, 2004). Total losses of livestock forage value due to C. solstitialis infestations on private land for the state of California were estimated at $9.45 million per year (Eagle et al., 2007). Although data on other invaded regions is scarce, it is expected that economic impacts could also be significant.

Data collection

The available information on C. solstitialis was collected using Google Scholar and the Web of Science during 2019, using the search term: <“Centaurea solstitialis” or “C. solstitialis” and “ethnobotany” or “ethnobotanical” or “medicinal” or “chemistry” or “traditional uses”>. Thirty-one articles published between January 1978 to December 2018 pertaining to the chemistry, ethnobotany, pharmacology and toxicology of C. solstitialis were identified and reviewed. Although there is a possibility that some articles written in languages other than English may have been omitted by our search engine, our search criteria unified several data sources in a comparable manner, facilitating the access to unconnected studies to provide valuable emerging information as a result. The information retrieved from the papers (country and region of origin of the plants, common local name, category of use, parts of the plant used, specific uses, preparation, and the name of the authors of the studies) was compiled into a table (Table S1). Each line corresponds to a category of use mentioned in an article (1 or 2 lines per article), as some articles mention more than one type of use for this species. Information about the chemical volatile compounds was also summarized in a table (Table S2). A range of relative abundance (%) was calculated based on the articles that made this information available. This includes the relative abundances of the compound in all the regions where it has been found, ranging from the lowest value to the highest value found for each compound. Two articles from California (Beck, Smith & Merrill, 2008; Oster et al., 2015) were not considered in the “Range (%)” row due to a lack of information about this variable. One article from Algeria (Lograda et al., 2013) was excluded in the “Parts of the plant” row because this information was not available in the article.

Results

Economic importance

Traditional uses in the native range

Traditional uses of C. solstitialis were found almost exclusively in its native range. The plant is used for many purposes, which have been grouped into three major categories: medicinal, edible, and raw material. A total of 31 articles on the traditional uses of the species have been found in different countries of the Mediterranean and Western Asia (see Fig. 1).

Figure 1 Number of articles by geographical origin, within the native range.

Most of the reported uses are medicinal (Fig. 2), and include the treatment of (i) respiratory ailments (common colds in humans and animals); (ii) digestive ailments (dysentery, stomach and abdominal pain); (iii) viral infections (herpes); (iv) protozoa diseases (malaria); (v) lesions of the soft tissues and skin (mouth sore in humans and animals, boils and warts, skin rash); (vi) eye conditions or (vii) or urolithiasis (kidney stones). The plant is also used as antipyretic, stomach tonic and diuretic. All aerial parts of the plant are used as food in Italy, Iran, Saudi Arabia and Turkey, being included in soups, or fried with eggs, used in pastry or simply boiled (Guarrera & Lucia, 2007; Lentini & Venza, 2007; Licata et al., 2016; Geraci et al., 2018; Al-Sodany, Bazaid & Mosallam, 2013; Ertuğ, 2004; Akan et al., 2013; Kargioǧlu et al., 2008). Aerial parts are also dried and fed to sheep during winter. Moreover, its stems and branches are used to make brooms in Turkey (Kargioǧlu et al., 2008) (Table S1, Fig. 2).

Figure 2 Number of studies reporting different ethnobotanical uses across countries.

Honey making is not shown for the native range as it is common throughout.

Honey production

The value of C. solstitialis for honey production is well known in its native range, and it is listed as a plant species that produces unifloral honey in Europe (Persano Oddo et al., 2004). Interestingly, the extensive monocultures that the species forms has resulted in a significant use for honey production in the introduced ranges. This phenomenon has been well documented in California (Zouhar, 2002). It was calculated that 150,000 colonies of bees in California depended upon yellow starthistle for their primary source of pollen back in 1954 (Cordy, 1954), and in 1985 it was estimated to yield US$150,000 to US$200,000 per year (Maddox, Mayfield & Poritz, 1985). Although it is an economically important plant, it is believed that the movement of honeybee colonies by beekeepers may inadvertently assist the further spread of this plant in the North-American range, because the species is predominantly an outcrosser species and it relies on pollinators (mainly honey bees) to set seeds. In Argentina it has been observed that honey bees visit this species intensively for pollen collection, and that the honey made of C. solstitialis pollen contained a high level of protein (Andrada & Telleria, 2005). (Naab, Tamame & Caccavari, 2008) characterized the honey of C. solstitialis produced in Argentina as being white, with low pollen loads and with a pH varying from 3.19 to 4.06. There is also pollen from this species in the honey produced in south Australia (Somerville, 2005), and it has been rated 5 in a scale of 1-5 (Birtchnell & Gibson, 2008) for possessing a “very high quality” for honey production.

Phytochemical constituents and secondary metabolites

The interest of the scientific community in the chemistry of C. solstitialis began when it was proven to be causing a neurotoxic disease in horses in California. Many authors have been trying to identify and characterize the chemical profile of the species since 1954. Cassady & Hokanson (1978) were the first to identify the triterpene 3α, 16α-Dihydroxytaraxene-3-acetate. Several sesquiterpene lactones (repin, subluteolide, acroptilin, janerin and cynaropicrin) were identified in C. solstitialis by Merrill & Stevens (1985). Jakupovic et al. (1986) identified a guaianolide and a germacranolide (sesquiterpene lactones) and two bisabolone derivatives for the first time. Masso, Bertran & Adzet (1979) found phenolic compounds, flavonoids, tannins and terpenoid and phytosterol derivatives in C. solstitialis. Thiessen & Hope (1969) isolated the sesquiterpenic lactone solstitialin and revealed its structure and configuration for the first time.

The analysis of the essential oil of C. solstitialis carried out up to date through gas chromatography - mass spectrometry (GC/MS) has provided a complete list of volatile chemical compounds and their relative abundance. Buttery et al. (1986) found out that germacrene D was the major volatile constituent of the flower buds of C. solstitialis plants collected in California. Other studies carried out in California also found germacrene D in higher concentrations than other compounds (Beck, Smith & Merrill, 2008; Oster et al., 2015). Binder, Turner & Flath (1990) analyzed the constituents of three different parts of plants collected in Turkey and identified 62 compounds including 22 sesquiterpenes, 11 C13 polyacetylenes, 10 aldehydes, seven acyclic and one cyclic olefinic hydrocarbon, five alcohols, two ketones, one acid and one ester. Germacrene D was also the major compound in these plants. Esmaeili et al. (2006) analysed the essential oil of the aerial parts of C. solstitialis from Iran and found that it was composed of eight monoterpenes (16.5%), nine sesquiterpenes (39.3%) and one aliphatic acid (30.8%). The major compounds were hexadecanoic acid and caryophyllene oxide, followed by 1,8-cineole and caryophyllene. Senatore et al. (2008) analysed the volatile compounds of C. solstitialis ssp. schouwii from Italy and found that the main compounds were caryophyllene and caryophyllene oxide. Carev et al. (2017) analysed the essential oil of the aerial parts of C. solstitialis from Croatia. The main compounds were nonoxigenated sesquiterpenes (23.8%), with germacrene D the dominant one, followed by longifolen (3.6%) and b-caryo-phyllene (1.6%). Aliphatic acids were the most abundant among nonterpene components, representing 44.4% of the total oil. Lograda et al. (2013) found 41 compounds in plants collected in Algeria, being the most represented n-heneicosane (17.30%), hexadecanoic acid (12.79%), n-tricosane (10.51%), n-pentacosane (5.64%) and caryophyllene oxide (5.03%).

Sotes et al. (2015) focused on the leaf surface chemistry, which represent the first line of plant defense against herbivores and analyzed the epicuticular chemistry of plants originating from native and non-native regions. A high amount of sesquiterpene lactones were found, but the epicuticular chemistry showed variation among regions, suggesting that the plant changes its chemistry according to the demanding of the environment. Three sesquiterpene lactones were identified for the first time in C. solstitialis: epoxyrepdiolide derivative, solstitialin A-3 13 diacetate and linichlorin A. In a more recent study, Irimia et al. (2019) applied the same methodology as Sotes et al. (2015), but analyzed more regions to have a more complete overview of the inter-regional variations. These authors also observed that the plants from the non-native range were more allelopathic, inhibiting the germination of seeds of other species significantly more than plants from the native range, which was consistent with the novel weapons hypothesis (Callaway & Ridenour, 2004).

A total of seven articles revealing the chemical compounds of C. solstitialis and their relative abundance (%) were found. Despite some differences in the methodology used to obtain the plant extracts and to perform the chromatographic analysis, these data were put together and compiled in a table to systematize all the chemical compounds that have ever been identified in C. solstitialis plants around the globe (Table S2). These studies have been carried out using plants from the native range (Turkey, Croatia, Italy, Iran and Spain) and from the non-native range (California, Argentina, Australia and Chile). Different parts of the plant have been analyzed, including leaves, stems, flower heads, flower buds and aerial parts in general. To obtain the oil most of the studies grinded the plant parts to identify all the compounds present in the plants, while two studies (Sotes et al., 2015; Irimia et al., 2019) analyzed only the leaf surface chemicals without damaging the leaves. A total of 161 compounds have been recorded in some part of the plant, with 108 only present in plants from the native range. Among these compounds, 44 were found only in Turkey. Only seven compounds were found exclusively in the non-native range, two terpene compounds: cynaropicrin 3-acetate, cynaropicrin 4′-acetate; and 5 nonterpene compounds: (E)-β-ocimene, (Z)-3-hexeno, (Z)-3-hexenyl propionate, 2-methoxytoluene, perillene. The fact that most unique compounds were found in Turkey (Figs. 3 and 4) is supportive of this region as the center of speciation of the taxon, and suggests that this region could possess the largest genetic and functional diversity for the species. This is in agreement with the results obtained by Eriksen et al. (2014), which revealed great heterogeneity for gene diversity, allelic richness and private allele values among populations in Eurasia, with plant populations from Turkey scoring the highest levels of genetic diversity.

The compounds which are present in higher concentrations (over 20% per sample) are repin, reaching the highest abundance in Chile; subluteolide with higher abundance in Australia; hexadecanoic acid and caryophyllene oxide, both reaching the higher concentrations in Iran. These are followed by janerin, epoxyrepdiolide, α-Linolenic acid, n-heinecosane and germacrene D (15%–20%). Six of these compounds are sesquiterpenes.

The most geographically transversal compound, found in eight of the nine countries, was heptacosane. The terpene compounds found in a higher variety of countries were the pentacyclic triterpenoidsα-amyrin, β-amyrin and taraxasterol, and the sesquiterpene lactones solstitialin A-13 acetate, acroptilin, epoxyrepdiolide, janerin, repin and subluteolide. Plants from the native range (Algeria, Croatia, Italy, Turkey) tend to have higher amounts of nonterpene in relation to terpene compounds. The opposite is observed in non-native ranges with California as the region with a higher diversity of terpenes (Fig. 5).

Figure 3 Number of compounds that have been identified exclusively in one region.

Figure 4 Number of compounds found exclusively in the native range.

Figure 5 Diversity of terpene and nonterpene compounds per region.

Pharmacology

Antioxidant

Şen et al. (2013) found out that the methanolic extracts of capitula and aerial parts of the C. solstitialis had good ability to scavenge free radicals despite having small amounts of phenolic compounds. Koc et al. (2015) went further and tested C. solstitialis for its potential medicinal action of biological targets that are participating in the antioxidant defense system such as catalase (CAT), glutathione S-transferase (GST), and glutathione peroxidase (GPx). The results showed high GPx and GST enzyme inhibition activity with acetone extracts from the flower of C. solstitialis, with IC50 (half maximal inhibitory concentration) values of 79 and 232 ng/mL, respectively.

Antiulcerogenic

Centaurea solstitialis has been used in the Turkish culture for many years to treat ulcers and stomach related diseases. In 1993, Yeşilada et al. (1993) based on ethnobotanical data, tested this species for its antiulcerogenic activity, and showed that the chloroform fraction of C. solstitialis exerts remarkable anti-Helicobacter pylori activity against both standard strain and clinical isolates at very low concentrations. H. pylori is a bacteria which causes ulcers, gastritis and cancer (Covacci et al., 1999).

The sesquiterpene lactones have been identified as the active constituents of the chloroform extract of the flowering aerial parts of the plant (especially chlorojanerin and 13-acetyl solstitialin A), and have been isolated through bioassay-guided fractionation procedures (Yesilada et al., 2004). A more recent study has revealed that each of the active compounds possesses a different anti-ulcer activity profile that interacts together in the plant remedy and show a remarkable effect (Gürbüz & Yesilada, 2007).

Antiviral and antimicrobial

Centaurea solstitialis has been tested for antimicrobial activity and has shown high activity against Staphylococcus aureus at a 0.5 mg/ml concentration. Therefore, C. solstitialis may be used as an antibiotic for S. aureus infections (Tekeli et al., 2011). Lograda et al. (2013) tested the biological activity of the essential oil of C. solstitialis grown in Algeria against nine bacterial strains, and it showed moderate to significant antibacterial activity.

The sesquiterpenic lactones centaurepensin, chlorojanerin and 13-acetyl solstitialin have been found to accelerate the healing process of labial and genital herpes lesions, providing scientific support for the utilization of C. solstitialis against herpes labialis infections in infants in Turkish folk medicine (Özçelik et al., 2009).

Antinociceptive and antipyretic

Akkol et al. (2009) obtained ethanol and aqueous extracts from the aerial parts and roots of C. solstitialis and tested it for antinociceptive effects using p-benzoquinone-induced writhing model in mice as a common in vivo activity assessment model. The ethanol extracts obtained from both aerial parts and roots showed significant antinociceptive activity, but the activity of the aerial parts was more prominent and close to that of the reference compound acetyl salicylic acid. Hexane and chloroform fractions exerted a potent antinociceptive activity, while n-butanol and remaining aqueous fractions were not significantly active. The ethanol extract of the aerial part also demonstrated a potent antipyretic activity, although less potent than acetyl salicylic acid.

Antiproliferative

Erenler et al. (2016) isolated two sesquiterpene lactones, solstitialin A and 15-dechloro-15-hydroxychlorojanerin, from the methanol extract of C. solstitialis stem and studied the anticancer activities of both compounds. The compounds exhibited significant anticancer activities against HeLa (Human uterus carcinoma) and C6 (Rat Brain tumor) cell lines in different concentrations. The stem extract was preferred for bioassay-guided isolation due to the highest activity. High activity was recorded even in lower concentrations (from 75 µg/mL to 5 µg/mL) for C6 cell lines. However, solstitialin A exhibited low activity at the concentration of 30 µg/mL against HeLa cell lines and did not show any activity at lower concentrations of 20, 10 and 5 µg/mL.

Toxicity studies

The first study on the toxicity of C. solstitialis was carried out in 1954, triggered by the emergence of a disease affecting horses in central and northern California, locally known as “chewing disease” or “yellow star thistle poisoning”, identified by scientists as “nigropallidal encephalomalacia”. The symptoms were abnormal movement disorders which resemble those of Parkinson’s disease in humans. It was demonstrated that this disease is linked to the ingestion of large amounts of C. solstitialis (Cordy, 1954). Aqueous-ethanolic extracts of the plant have been proven to be toxic to rats, mice and monkeys in moderate dosages (Mettler & Stern, 1963)).

Some authors have identified and isolated (through a bioactivity-guided fractionation approach) some neurotoxic sesquiterpenoids from C. solstitialis which may be responsible for causing the disease in horses. Cassady et al. (1979) identified centaurepsin as a cytotoxic constituent. Stevens, Riopelle & Wong (1990) isolated repin from C. solstitialis plants, which is considered to be the major neurotoxic compound. Wang et al. (1991) found out that, among the compounds isolated during the study, 13-0-acetylsolstitialin A and cynaropicrin exhibited neurotoxic activity against cultured rat foetal brain cells depending on the concentration. These results have also been supported by Cheng et al. (1992). Hay et al. (1994) showed that the toxicity of these sesquiterpene lactones is due to the reactiveα-methylene function. Roy, Peyton & Spencer (1995) isolated and characterized aspartic acid and glutamic acid as two potent neuroexcitotoxic compounds, being aspartic acid the main toxic component in the alcoholic extract of the plant.

Moret et al. (2005) obtained a complete profile of the free nitrogenous fraction of C. solstitialis through HPLC procedures and found no particularly high amounts of excitotoxic amino acids in polar extracts of the plant. Tyramine was identified as the most important biologically active amine present in C. solstitialis, and the authors suggest that the prolonged consumption of the tyramine containing plant may be, at least partially, responsible for toxic effects observed in horses, but further investigation is needed.

Conclusions

The ethnobotanical literature available for the model invasive weed yellow star-thistle showed a diversified range of traditional uses including medicinal, gastronomic, and as prime material, conferring an important economic and cultural value to the species in its native range. However, the only confirmed use of the species in the non-native range was honey-making and, indirectly, as forage, but only within the context of planned weed-control interventions.

Traditional knowledge is the consequence of in-situ experimentation, usually for millennia (Tempesta & King, 1994) and significant human migrations are usually accompanied not only by the introduction of useful plants, but also by the knowledge on how to use them (De Medeiros et al., 2011). Of the numerous traditional uses of C. solstitialis in its native range the medicinal uses are the most representative, with 16 different specific uses for a range of medical procedures and conditions, including as antiseptic. Interestingly for a plant considered to be medicinal, the species is also considered a culinary ingredient across several countries of the native range. However, more than half of the ethnobotanical studies which mention C. solstitialis had been carried out in Turkey, its ancestral range and its center of speciation, and thus where the species has been historically present for the longest time. Other countries in what is considered the “expanded” native range of the species across the Western Mediterranean, including Italy, have fewer records of medicinal uses even though, curiously, there were more studies reporting its use as a food ingredient in Italy than in Turkey. This exemplifies how the number of studies, per se, might be an imperfect indicator of actual use, as the choice of what to study must be biased by regional differences in cultural interests. Regardless, we observed a gradient within the native range with numerous and diverse ethnobotanical uses in the ancestral native range of the Eastern Mediterranean and Western Asia, where the species first originated, and gradually less frequent uses as we move towards the expanded native range on the Western Mediterranean. Medicinal uses were particularly slow to be transmitted througout the expanded native range, with most studies of such kind concentrated in the ancestral range of the Mediterranean west, and gradually less reports as we go east, with no uses reported for e.g., Spain, where it is also considered a native weed. The absence of reported ethnobotanical uses in Spain could be a main driver of the lack of ethnobotanical uses in the Americas, as American C. solstitialis populations originated predominantly from Spain, at least initially (Eriksen et al., 2014; Barker et al., 2017) and Hispanic culture is prevalent in the South (Argentina, Chile) and North American (California) regions where the species was first introduced. This supports the idea that availability of a potentially useful plant—availability hypothesis—is a necessary condition for ethnobotanical adoption, although rarely a determinant of it (Hart et al., 2017; Soldati et al., 2017). The same reasons that prevented the species from being introduced into Western Europe pharmacopeas, in spite of plant availability and close cultural connections, could also be at play in the non-native range of the species. We can only speculate about the actual reasons, but it could be due to the presence of other plants already providing with the same medicinal properties —diversification hypothesis—making it unnecessary if those other species are also abundant (Hart et al., 2017).

The lack of transmission of cultural knowledge to the non-native regions of the species is in striking difference with the well documented transmission of ethnobotanical knowledge across continents during significant human migrations (Pfeiffer & Voeks, 2008; De Medeiros et al., 2011). For instance during the European colonization of the Americas, abundant ethnobotanical knowledge was brought from West Africa and the Mediterranean, when migrants either brought with them both plants of interest and the knowledge of how to use them, or were able to find substitutes with similar uses in the new colonies (Voeks & Rashford, 2012; Moret, 2013). This has also been documented in reverse, and Colombian migrants have been documented to bring ethnobotanical remedies from America into the UK (Ceuterick et al., 2008). In contrast, our work shows how biological introductions which are not paralleled by significant human migrations can result in a predictably negligible cultural transmission, but also on a very slow local discovery and development of cultural uses—notice that C. solstitialis was accidentally introduced into the Americas less than 200 years ago, long after Europeans were already well established there. Acknowledgely, we might have missed cultural uses that are not reported in scientific literature, but our methodology was applied coherently among the native and non-native ranges of the species, and there is no reason to expect that any of the studied regions would have a larger amount of scientific literature. If anything, we could expect more studies in the USA, where we could not find any use beyond honey making. Interestingly, even within the native range of the species, different types of ethnobotanical knowledge were transmitted at significantly different rhythms, being particularly slow for medicinal uses and possibly slightly faster for culinary uses. Plant invasions are unplanned experiments that allow us to study the ecological and evolutionary processes unfolding during the colonization of new regions (Hierro, Maron & Callaway, 2005; Irimia et al., 2021; Montesinos, 2022), our results show how they can also be used as models that allow us to understand, in real time, how ethnobotanical culture is created and transmitted.

Pharmacological studies have provided support to most of the medicinal uses of our target species, confirming that the species contains chemicals that possess antiviral, antimicrobial, antipyretic, antinociceptive, antiulcerogenic, antioxidant, and antiproliferative properties, and that plants from the native range present a richer variety of pharmaceutically active compounds than plants from the non-native range. Invasive plants frequently use active chemical compounds as chemical defenses against predators, herbivores, and pathogens which are expected to be more abundant in the native than in the non-native range (Liu & Stiling, 2006; Correia et al., 2016). These defenses can be quantitative (digestibly reducers) to deter specialist herbivores, or qualitative (toxins) to deter generalists (Müller-Schärer, Schaffner & Steinger, 2004). Qualitative chemical defenses (frequently alkaloids) are the ones conferring plants most medicinal properties, but the amount of these chemicals is dependent on genetic and environmental factors, and are known to vary geographically (Sotes et al., 2015; Irimia et al., 2019). The Shifting Defense Hypothesis (Joshi & Vrieling, 2005) poses that when an exotic plant is introduced into a new region where specialist herbivores are frequently absent, plants experience selective pressures to increase the amount of qualitative defenses in these non-native regions (e.g., alkaloids). This directly links with the disproportionate success that these chemical defenses, which might be new to the recipient communities, confer to some invasive species, in what is known as the Novel Weapons Hypothesis (Callaway & Ridenour, 2004). Studies with our model species suggest that novel weapons might contribute to its success in the regions that they invade, but also provide evidence for higher concentration of qualitative defenses in the non-native range of the species, in the form of pharmaceutically active sesquiterpene lactones, paralleled by a reduction in quantitative defenses (Sotes et al., 2015). Thus, on one side we find a richer chemical diversity in the native range of the species, which might contribute to explain the abundant ethnobotanical uses described there, but on the other hand the concentration of pharmaceutically active compounds is higher in at least some non-native regions, which shows potential for ethnobotanical uses yet to be discovered in these invaded areas. Within the native range, we did observe a decrease in both chemical richness and reported ethnobotanical uses as we went from the Mediterranean west to the east, however, this could be a confounding factor that does not necessarily imply that ethnobotanical uses are less frequent because of a lower chemical diversity, since a shorter historical exposure to the plant could also be playing an important role. Our review highlights both the importance of chemical biogeography and the long times involved in the discovery and transmission of cultural plant uses.

Overall, our review exemplifies the usefulness of reviews of the ethnobotanic literature about specific invasive taxa. The ancestral range of the invasive weed C. solstitialis was where the most numerous and diverse ethnobotanical uses had been described, and are also the regions holding the highest chemical and functional diversity. In the non-native regions the species over-abundance is resulting in significant environmental and economic problems, but also in some incipient economic and cultural activity, such as honey production. As an emerging insight, our work showcases the slow process of cultural integration of exotic species into daily uses, particularly when biological introductions are not accompanied by significant human migrations.

Supplemental Information

Supplemental Information 1 Traditional uses of C. solstitialis organized by country, common name, category of use, part of the plant used, specific uses, preparation and the author of the respective study

Click here for additional data file.

Supplemental Information 2 List of volatile compounds identified in C. solstitialis plants, organized by type of compound (terpene or nonterpene), relative abundance range, region of origin, parts of the plant analysed and the authors of the respective study

Click here for additional data file.

Additional Information and Declarations

Competing Interests

Author Contributions

Data Availability

The authors declare there are no competing interests.

Soraia Branco conceived and designed the experiments, performed the experiments, analyzed the data, prepared figures and/or tables, authored or reviewed drafts of the article, and approved the final draft.

Ramona E. Irimia analyzed the data, authored or reviewed drafts of the article, and approved the final draft.

Daniel Montesinos conceived and designed the experiments, analyzed the data, authored or reviewed drafts of the article, and approved the final draft.

The following information was supplied regarding data availability:

This is a literature review.

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
