# Peer review of "The introduction of an invasive weed was not followed by the introduction of ethnobotanical knowledge: a review on the ethnobotany of Centaurea solstitialis L. (Asteraceae)"

_PeerJ, doi:10.7717/peerj.15489_

## Round 0.1 · original submission · Major Revisions

We received two reviews which both found your article interesting and valuable, but also pointed to some issues that would need to be addressed prior to acceptance. The two reviews contain detailed suggestions for improvement, which I strongly encourage you to consider for your revision. While I can see that your literature review was not based on a simple google search (as indicated by Reviewer #2), I still agree with Reviewer #2 concerning the need for more references. Most importantly, the findings presented for the focal species (Centaurea solstitialis) should be discussed in the light of other ethnobotanical work on the use of alien species in their invaded ranges. Reviewer #2 provides a long list of respective papers, which may also help to address points 3 (and possibly also 4) raised by Reviewer #1.

·

Basic reporting

The manuscript describes an analysis about whether the introduction of a species to its non-native range is followed by ethnobotanical uses in a scenario in which the introduction of this species is not accompanied by massive human migrations. They conducted this analysis by reviewing academic studies about the species in different parts of the world. They concluded that, in this scenario, the introduction of this species is not followed by ethnobotanical uses.

The article is well written regarding the English language. It is written in a professional structure and all the data to support their conclusions is shared.

I do think there are some minor issues with the clarity as described below:

Lines 39-41: This first phrase is not very clear. It begins stating that the invasive species break havoc (which is a statement about the natural world), and then proceeds to contrast with how they are “considered” [by someone] to be in their native habitat. If the authors agree with both statements, I think both should be written in a more direct form. For example, “...but in their native range they are an inconvenient but not highly problematic native weed”. Or reversely, both statements should be referring to a discussion they are not taking part in yet. “Invasive species are usually considered to cause havoc…”

Lines 47-50: It is conceptually misleading to say that “exposure to chemicals for extended periods of time” can be considered an evolutionary arms race, because evolutionary arms race is a complex eco-evolutive process. It would be best to rephrase it to pass on the idea that the evolution of such chemicals in a region allows other species of the same region to adapt to it.

Lines 65-67: The relationship established in this sentence is not clear. For me it is not clear why the possibility of substitution of an overexploited native plant for an invasive one would be a scenario particularly helpful in the identification of pharmaceutical research targets.

Lines 67-68: I’m assuming that this means that, knowing the pharmaceutical potential of invasive species, one could encourage the human communities to make more use of it and therefore limit more effectively its population growth. Good point, but this argument needs to be more explicit here.

Line 70: a closing parenthesis is missing.

Line 92: “species can provide with important information” the “with” seems to be out of place

Lines: 239, 256, 267, 275, 298, 307, 334, : Wrong use of parentheses in text citation.

Lines 347-348: This is a strong statement and needs citation.

Experimental design

I think the study design is adequate and well presented.

While I was reading I was particularly concerned about the implication that ethnobotanical studies were used as an index of ethnobotanical studies. However, I think the authors presented a good argument of how these things are related. In the "Validity of the findings" section, I suggest a way to make this argument easier to present.

Validity of the findings

This is a very important study. The data the authors present shows relevant evidence connecting introducing species to its non-native range and how human societies react to these species. I think this is an important scientific analysis but I also think the presentation of the article, as it is, is not showing how great these findings are. Therefore, I would recommend reviewing before it is published.

My main concern is about how the article is presented. I think the article is lacking focus on what is really relevant to the discussion they are bringing. See my points and suggestions below. Please understand that my suggestions are intended to facilitate the authors' work and should not be seen as the only way to solve the issues I presented.

Point 1: Too much space is used to describe the phytochemical constituents, secondary metabolites and pharmacological components. In the context of the research question here, these are only important if they can be used to understand geographical ethnobotanical use patterns. For example, in the section “Pharmacology” it is not clear how this description of pharmacological uses helps the reader understand the geographical ethnobotanical use patterns because it does not refer to how these components are used or not in the native and non-native distribution of the plant. I assume that this section is supposed to show evidence that the plant has pharmacological uses, but the article is not about if the plant has or hasn't actual uses. It's about if people use it or not.
Suggestion 1: Only describe components details when these details help the reader understand why its use is present or not present in a region. If the objective is to list all potential uses of the plant that are not present in one or more regions, a table should do.

Point 2: The figures used are not the best way to present the data. I do think the data necessary to evidentiate the conclusions are there, but to understand the main pattern being presented I need to merge all these pie and bar charts in my head which makes it difficult to visualise the evidence.
Suggestion 2: Maybe a regional or world map showing the distribution of the plant, its native and non-native habitats and overlays of the ethnobotanical uses could make evident that the distribution of the species is not followed by the distribution of uses and how this changes from type of use to type of use. This would also help us understand the dispersion of uses in the native and ancestral range. It would also be useful to have one of these maps showing total number of studies produced in the region to help you better address the "ethnobotanical uses X ethnobotanical studies" issue.

Point 3: The “Conclusions” section is not pointing towards ethnobotanical explanations enough. The main point of the article is that the introduction of the species in some parts of the world is not followed by its potential ethnobotanical uses. They do not propose an ethnological explanation for it.
Suggestion 3: I think the discussion would be more rich if it contrasted different uses and the three ranges of distribution they explored (native, non-native, ancient). For example, the authors have said that medical uses are hard to pass on. Why is that? Is it due to the nature of medical uses or because the maladies they treat are not common in other parts of the world? Are there other papers that discuss how difficult it is to pass medical uses? What about the other uses, are they difficult to pass also? The lack of ethnobotanical use in other regions could be explained by the specific way in which this plant was introduced? Could the lack of uses be explained by how recent this introduction was or there is another barrier to the passage of uses that would prevent this passage indefinitely. I think maybe understanding the relationship between the ancient and modern distributions could be an interesting way to shed light on how and why some uses are present or not in the world.

Point 4: It lacks a discussion about the point brought in the introduction about control of introduced species.
Suggestion: Describe how the knowledge you generated here can be used to foster better actions of species control. If it is not possible to do that, this point should not be mentioned in the introduction.

Additional comments

I hope the points I made will help further develop this work. I look forward to read the new version of this paper and how the authors decided to deal with the issues pointed.

Reviewer 2 ·

Basic reporting

I have reviewed the manuscript “The introduction of an invasive weed was not followed by the introduction of ethnobotanical knowledge: A review on the ethnobotany of Centaurea solstitialis L. (Asteraceae)”. My overall assessment is that it is a nice review of the chemistry and uses of the species in its native range. There are some minor problems with the English that copy edit can easily fix.
However, the authors go on to make conclusions related to immigration and ethnobotanical transfer that are completely unfounded. They refer, for example, to a lack of human immigration to these alien landscapes as a cause of lack of use, but fail to include a single reference on human migration. As a California resident, I can state that there are almost no nationalities not well represented in our state of 40 million people. Why the authors felt inclined to make statements about immigration without actually exploring the issue, and without providing any reference,s is perplexing.
The second and major problem with this manuscript is that there is in fact a significant ethnobotanical literature dealing with the theoretical issue of alien species and their incorporation (or lack thereof) into local plant pharmacopeias. But the authors have included just a single reference (Bennett and Prance) on this. This is mostly because they simply Googled the species name, and gleaned what they could from articles dealing with starthistle. I am listing below a brief list of ethnobotanical research that addresses these questions, both in a cultural, historical, and biochemical sense. Check out in particular Albuquerque 2006 (diversification hypothesis) and Gaoue 2017 for related ethnobotanical theories. There are in addition numerous examples of research exploring the outcome of specific plants invading new areas of the globe (see for example several studies by Dan Austin), and how it was or was not incorporated into local plant uses. Again, a simple Goggle sweep of references just related to starthistle would not reveal these sources.
Finally, as someone quite familiar with starthistle in California, I can state that it is perhaps the most unpleasant invasive species we have. It has the worst thorns of almost any plant in California—native or exotic. It is not a surprise to me at all that people have not incorporated it into their repertoire of useful species.
My suggestion is that either the authors remove all reference to the question of the use of this invasive species in non-native lands, and just focus on the various properties and chemistry in its native ranges; that really is what most of the article is about anyway. Or, if they choose to pursue the topic of ethnobotanical acquisition in newly invaded landscapes, that they do a deep dive into the literature. It is an interesting topic, and the authors may well come up with some novel conclusions.
See the following, as well as references listed by these authors:
Albuquerque, U.P., 2006. Re-examining hypotheses concerning the use and knowledge of medicinal plants: a study in the Caatinga vegetation of NE Brazil. Journal of ethnobiology and ethnomedicine, 2(1), pp.1-10.
Alm, T., 2013. Ethnobotany of Heracleum persicum Desf. ex Fisch., an invasive species in Norway, or how plant names, uses, and other traditions evolve. Journal of ethnobiology and ethnomedicine, 9(1), pp.1-13.
Atyosi, Z., Ramarumo, L.J. and Maroyi, A. (2019). Alien plants in the Eastern Cape province in South Africa: Perceptions of their contributions to livelihoods of local communities. Sustainability ,11(18), 5043
Austin, D.F., 2013. Moon–flower (Ipomoea alba, Convolvulaceae)—medicine, rubber enabler, and ornamental: a review. Economic botany, 67(3), pp.244-262.
Austin, D.F., 2014. Salt marsh morning-glory (Ipomoea sagittata, Convolvulaceae)—An amphi-Atlantic species. Economic botany, 68(2), pp.203-219.
Gaoue, O.G., Coe, M.A., Bond, M., Hart, G., Seyler, B.C. and McMillen, H., 2017. Theories and major hypotheses in ethnobotany. Economic Botany, 71(3), pp.269-287.
Maema, L.P., Potgieter, M.J. and Samie, A., 2019. Ethnobotanical survey of invasive alien plant species used in the treatment of sexually transmitted infections in Waterberg district, South Africa. South African Journal of Botany, 122, pp.391-400.
Maldonado, G. and Voeks, R. 2021. The Paradox of Culturally Useful Invasive Species: Southern Cattail (Typha domingensis) Crafts of Lake Patzcuaro, Mexico. Journal of Latin American Geography. 20: 148-174.
Medeiros PMD, Soldati GT, Alencar NL, Vandebroek I, Pieroni A, Hanazaki N, Albuquerque UP (2012) The use of medicinal plants by migrant people: adaptation, maintenance, and replacement. Evid Comp Alt Med, Article ID 807452, 11 pages, http://dx.doi.org/10.1155/2012/807452
Pfeiffer, J. and Voeks, R. 2008. Biological invasions and biocultural diversity: Linking ecological and cultural systems. Environmental Conservation 35 (4): 281-293.
Rakotoarisoa, T. F., Richter, T., Rakotondramanana, H., and Mantilla-Contreras, J. (2016). Turning a problem into profit: Using Water Hyacinth (Eichhornia crassipes) for making handicrafts at Lake Alaotra, Madagascar. Economic Botany, 70(4), 365-379.
Santos, L., Nascimento, A., Vieira, F., da Silva, A., Voeks, R., and U. Albuquerque. 2014. The Cultural Value of Invasive Species: A Case Study from Semi-arid Northeastern Brazil. Economic Botany 68 (3): 283-300.
Shackleton, S.E. and Shackleton, R.T. (2018). Local knowledge regarding ecosystem services and disservices from invasive alien plants in the arid Kalahari, South Africa. Journal of Arid Environments, 159, 22-33.
Soldati, G.T., de Medeiros, P.M., Duque-Brasil, R., Coelho, F.M.G. and Albuquerque, U.P., 2017. How do people select plants for use? Matching the ecological apparency hypothesis with optimal foraging theory. Environment, Development and Sustainability, 19(6), pp.2143-2161.

Experimental design

See above.

Validity of the findings

See above.

---

## Round 0.2 · Minor Revisions

Dear Authors,

Thank you for this revision. One of the previous reviewers agreed to review this revised version, and was quite satisfied with your revision. I agree with this assessment.

The reviewer points out, however, that pie charts are not ideal for representing data. In line with this point of critique, I would like to ask you to revise Figs. 1, 3 and 4 accordingly.

One more minor comment: In Line 470, shouldn't it read "These defenses can be quantitative (digestibly reducers) " instead of "These defenses can be qualitative quantitative (digestibly reducers) "?

I am looking forward to the final version!

Kind regards,
Tina Heger

·

Basic reporting

I believe the authors have adequately addressed the issues I pointed out in the first version. Regarding science, I do not see any more problems with the paper.
- I still think better figures would do good for the article's visibility and would help the authors convey the message more clearly. I understand that creating a map would be difficult, however, I would strongly recommend that they change the piecharts for bar charts. Piecharts are disrecommended for this kind of data because it is difficult for our brains to pick on the differences in the area of each of the slices of the pie. This is clear in this example: https://en.wikipedia.org/wiki/Misleading_graph

Authors may argue that the value of each slice is also printed in the figure. This makes me wonder, if what I am going to use to understand the chart are the numbers, why is there a chart there? The objective of the charts is to make evident a general pattern in the general data. I'm afraid the pie chart is not helping much with this.

I would leave this decision in the hands of the authors, however.

Experimental design

No comment.

Validity of the findings

No comment.

---

## Round 0.3 · accepted · Accept

Dear Authors,
Thank you for this revision, especially for revising the figures. I think that now, all issues are resolved, and I am happy to accept the current version for publication.
Kind regards,
Tina Heger